# Acceptability and implementation potential of a health literacy intervention to increase colorectal cancer screening in deprived areas: A qualitative study of patients and general practitioners participating in a cluster randomized controlled trial

Alix Boirot[1*], Maria Claudia Addamiano[2], Clémence Casanova[1],
Niamh M. Redmond[2,3], Géraldine Cazorla[1], Michel Rotily[4],
Anne-Marie Schott-Petelaz[5], Christian Balamou[6], Zineb Doukhi[7],
Myriam Kaou[5], Françoise Couranjou[8], Julien Mancini[1,9], Cyrille Delpierre[2],
Marie-Anne Durand[10,11], Aurore Lamouroux[12,13]

1 Aix Marseille Univ, Inserm, IRD, ISSPAM, SESSTIM, Sciences Economiques & Sociales de la Santé & Traitement de l'Information Médicale, Marseille, France, 2 EQUITY research team (Certified by the French League Against Cancer), CERPOP, UMR, Inserm, Université Toulouse III Paul Sabatier, France, 3 ARC West/Bristol Medical School, University of Bristol, United Kingdom, 4 Centre d'Etudes et de Recherche sur les services de santé et la qualité de vie (CEReSS), Aix Marseille Université, Marseille, France, 5 Research on Healthcare Performance (RESHAPE) INSERM U1290, Université Claude Bernard Lyon 1, Lyon, France, 6 Centre Régional de Coordination du Dépistage des Cancers en Auvergne-Rhône-Alpes - CRCDC-AuRA, Site de l'Ain, Bourg-en-Bresse, France, 7 DMU PRISME Délégation à la Recherche Clinique et à l'innovation (DRCI), Unité d'épidémiologie Clinique/ CIC-EC 1426 – INSERM, France, 8 Assistance Publique des Hôpitaux de Marseille (APHM), Direction de la Recherche en santé, Marseille, France, 9 APHM, Hop Timone, BioSTIC, Biostatistique et Technologies de l'Information et de la Communication, Marseille, France, 10 The Dartmouth Institute for Health Policy & Clinical Practice, Dartmouth College, Lebanon, New Hampshire, United States of America, 11 Unisanté, University Center for General Medicine and Public Health, Switzerland, 12 APHM, Centre de Santé Universitaire – Espace Santé Aygalades, Marseille, France, 13 Comité Départemental d'Education pour la Santé (CoDES 84), Avignon, France

☉ These authors contributed equally to this work.
* alix.boirot@univ-amu.fr

## Abstract

### Background

Colorectal cancer (CCR) is one of the leading causes of death worldwide. Early detection remains a highly effective strategy for curing this disease. In France, despite a free organised screening programme for people aged between 50 and 74, participation rates remain suboptimal. Socioeconomic position and health literacy levels exacerbate the situation, with the lowest screening rates observed in the most socially disadvantaged areas. This study assessed patients' and General Practitioners' (GP) views on the acceptability of an intervention to increase screening uptake using a simple brochure and video on the importance and process of CCR screening.

**Data availability statement:** The raw data for this study consist of interview transcripts containing potentially identifying information. As participants did not consent to the sharing of their data beyond the study team, these data cannot be made publicly available. However, relevant de-identified excerpts from the transcripts are included in the article. In accordance with the trial's ethical approval and participant consent procedures, data will be available upon request and is subject to approval by the Trial Steering Committee, a supplementary approval by the University of Toulouse Ethics Committee, and a data sharing agreement. For further information or data requests, please contact either the CERPOP department administrative and financial manager, Sandra Bourgouin (cerpop.gestion@inserm.fr), the corresponding author, Dr. Alix Boirot (alix.boirot@univ-amu.fr) or the trial manager, Dr. Niamh M. Redmond (niamh-maria.redmond@univ-tlse3.fr).

**Funding:** This research was supported by a grant (2020-006) from French National Cancer Institute (INCa) https://en.e-cancer.fr/. The funders had no role in study design, data collection and analysis, decision to publish, or preparation of the manuscript.

**Competing interests:** Marie-Anne Durand has contributed to the development of Option Grid patient decision aids. EBSCO Information Services sells subscription access to Option Grid patient decision aids. She receives consulting income from EBSCO Health and royalties. This does not alter our adherence to PLOS ONE policies on sharing data and materials. No other competing interests declared.

## Method

We conducted a cross-sectional qualitative study using semi-structured interviews with patients (n = 24) and GPs (n = 22) who used or participated in the DECODE project intervention. The interviews were conducted by telephone or videoconference and analysed thematically using Nvivo software and dual independent coding.

## Results

95% of GPs expressed a clear preference for the video over the brochure. Patients had varied results with 50% preferring the video, as it demonstrated how to do the test, versus the brochure. The humorous and de-dramatising aspects of the video were the two key factors highlighted by interviewees. However, support from healthcare staff (GPs, nurses, etc.) is still essential, in supporting patients in prevention. This presents a challenge for GPs, who are frequently constrained by time limitations during consultations.

## Conclusion

Our findings emphasize the need to tailor promotional materials for both patients and healthcare professionals to improve CCR screening uptake, balancing digital efficiency with maintaining core human relationships in healthcare. Such intervention can be integrated into different workflows. The addition of video into national CRC screening programs might also help. Targeting CRC screening interventions at provider-patient interactions, ensuring they are tailored, accessible, and engaging, is key to reducing disparities.

## Introduction

Colorectal cancer (CRC) is one of the leading causes of death worldwide [1–3]. According to the World Health Organization (WHO), since 2020, 1.8 million new cases of colorectal cancer have been recorded worldwide, making it the third most common cancer after lung and breast cancer, and the second leading cause of cancer deaths. This disease predominantly affects Western countries, with particularly high incidence rates observed in Europe, Australia and New Zealand [4]. CRC is often detected late and as a result, diagnosis is frequently too far advanced for effective treatment. Paradoxically, if diagnosed early, CRC is one of the easiest cancers to cure, with ablative surgery and associated chemotherapy treatments.

In France, a national CRC screening programme targeting individuals aged 50–74 has been in place since 2009. A free home screening kit is available to eligible individuals from their General Practitioner (GP) or a pharmacy via a letter to their home address. Since 2022, the screening kit is also available online. Extensive evidence, across cancer types, demonstrates the benefits of organized cancer screening programmes [5–9] in terms of reduced cancer incidence and relative morbidity. However, for a screening program to be effective, a 45% uptake rate is required [10]. National

data show that in 2018–2019, only 34.3% of eligible individuals undertook the screening test [11]. Since the implementation of the national CRC screening programme, participation rates have remained unchanged despite campaigns to increase accessibility to CRC screening kits – available for free in pharmacies or directly online at ameli.fr [12]. A recent study [13] demonstrated that in France the most disadvantaged populations (based on European Deprivation Index (EDI) data) had a lower net 5-year survival rate for most cancers and in particular for CRC. These findings align with international data [14]. Additionally, social inequalities in health affect the most disadvantaged populations both in terms of higher exposure to risk factors and reduced access to care [15]. As well as socioeconomic position, lower health literacy is associated with poor health outcomes, poorer overall health status, higher mortality rates [16], and less engagement with prevention services including CRC screening [17].

Given the socially graded uptake of CRC screening, health literacy (HL) has been shown to be an important determinant in participation [17–19]. HL can be defined as "the degree to which individuals have the capacity to obtain, process and understand basic health information and services needed to make appropriate health decisions" [20]. Associations between HL and CRC screening have been established. People with limited HL are less likely to seek and understand CRC screening information. Possible solutions include developing information tools using simple language, images, and good principles of clear communication [21]. The involvement of general practitioners (GPs) is essential, as they serve as the primary point of contact for many people. Our hypothesis is that the manner in which the importance of the CRC test is introduced and presented, using clear language plays a crucial role in screening rates, particularly among the most disadvantaged populations. Indeed, previous studies of screening interventions have highlighted the importance of public involvement in research and have addressed the development of interventions and strategies to improve participation in screening [22]. In light of this, we conducted a multi-center cluster-randomized controlled trial which aimed to increase participation in the organized screening programme: the DECODE trial [23] (NCT04631692). The trial, using a community-based participatory approach, emphasized the importance of considering HL of the target audience (patients) and the knowledge of patient HL issues by GPs.

Alongside the trial, we carried out a qualitative study with participating GPs and patients. This study aimed to assess the acceptability of the intervention among patients in GP practices enrolled in the trial and, more specifically, to explore the implementation and normalization of the intervention by evaluating the perceptions of both participating GPs and patients. We report here the results of our qualitative analysis.

## Materials and methods

The trial protocol details the methods [23]. Briefly, the trial aimed to determine whether a mixed health literacy intervention targeting both GPs and patients in disadvantaged areas in four French regions improved the screening uptake after 1 year compared to a usual care group.

### Ethics

Ethical approval for the study 2021−349 was obtained by the the University of Toulouse Ethics Committee (CER – Comité d'Éthique de la Recherche) on 8 of March 2021 and the DECODE trial was registered on ClinicalTrials.gov with the trial registration number 2020-A01687-32. The Ethics Committee approved a classification of 'verbal consent required' for the trial as the main trial was testing a combined health literacy training intervention for GPs and a patient leaflet and video for patients and deemed verbal consent appropriate. We assigned an inclusion number to each participant in the DECODE trial. This enabled us to link sociodemographic and literacy data to the qualitative data from the interview for patients who had verbally agreed to participate.

Of the 32 items of the COREQ checklist, item 23, which concerned the return of transcripts to participants, and item 28, concerning the verification of feedback by participants, were not applied in the process of collecting and analyzing the qualitative data of this study.

## Design

The present study is a cross-sectional qualitative study involving interviews with GPs and patients. GPs and patients were asked about the acceptability of the brochure and video aspect of the intervention, while GPs were also asked about the online training [24].

All methods and results are reported using the COnsolidated criteria for REporting Qualitative (COREQ) research checklist (Additional file 1 COREQ checklist).

## Brochure and video

The intervention for patients consisted of two components: 1) a pictorial brochure and 2) an online video produced by the French National Cancer Institute (INCa) that provided practical information on how to conduct the CRC screening test [25].

The printed brochure focused on the main steps of the CRC self-test but also delved into the significance of testing, the target population, and the optimal timing to undertake the test. The brochure was developed using a participatory approach, involving a community advisory group made up of patient experts, citizens, and representatives from cancer prevention institutions [26]. Particular attention was paid to the language used and how the messages were perceived, receiving feedback from individuals who were not at ease with the French language. Verbal consent to participate in the qualitative study was obtained by telephone with participants informed that they were free to stop the interview at any time and they had the right to ask that the interview not be recorded. In addition, we asked the participants to give verbal consent for the interviews to be recorded.

## Setting and participants

**General Practitioner recruitment.** GPs were invited to participate in the DECODE trial from the 4 regions involved. We used the 2015 European Deprivation Index (EDI) [27] to randomly select only GP practices with an EDI of 4 or 5 (most deprived). GPs were contacted by email or telephone by the research team and follow-up if necessary. We used our research and community contacts to reach other GPs via 'snowballing' to invite other GPs to participate. GPs who responded positively were informed about the trial processes and asked to complete a consent form, which included consent to participate in the qualitative study. GPs were block randomised into the control group or intervention group stratified by region and practice to avoid contamination.

**Patient recruitment.** Patients were recruited between October 2021 and April 2023 and were eligible to participate if they were aged between 50 and 74 years old, eligible for the national CRC screening programme (i.e., did not conduct a CRC screening test within the previous two years) and were able to complete a questionnaire in French alone or with help from a caregiver or relative, or interpreter.

GPs recruited patients during their routine consultations and were given a patient information leaflet explaining the trial itself and an invitation to participate in an interview study. As approved by CER, GPs obtained verbal consent from the patient, explained the study to them, and gave them a CRC self-test kit. In the intervention group, the GPs also used their training, the pictorial brochure, and the video to explain how to do the CRC screening test at home. All patients were followed up by telephone after 1 week and then 1 year to determine if they had taken the test (primary outcome).

**For the qualitative study.** At the final follow-up point, all patients in the intervention group were reminded of the opportunity to participate in an interview study via a standardized script. If they agreed to further information, the researcher conducting the qualitative interviews (CC) was informed of their interest and their contact details, socio-demographic data and responses to the standardized HL and health inequalities questionnaires were sent to ensure a balanced sample of patients subsequently contacted. Throughout the course of the interviews, the profiles of patients who had already completed the interview (region, age, sex) were reviewed, and the profiles of patients who were missing were highlighted to the Cras. This process was carried out until data saturation was reached. At each interaction with a

recruited patient, we checked they were still in agreement to consent, giving participants the opportunity to withdraw if they did not want to continue.

In May 2023 (post the trial recruitment period), all 32 intervention group GPs were reminded about the qualitative study and invited again to participate in an interview by two qualitative researchers.

## Data collection

Data were collected via semi-structured telephone and videoconference (via Zoom or Teams platforms) interviews, depending on the participants' preferences (both GPs and patients). The interview guides were developed based on Normalization Process Theory (NPT) [28], a theoretical model that assists in understanding the integration of complex interventions into standard healthcare practices. The interview guides (appendix 1) comprised 10 questions for patients and 21 for GPs, with follow-up questions if required. All Interviews lasted between 30 and 50 minutes. All data were collected between January 2023 and December 2023.

Interviews with GPs were performed by two researchers (AB and GC) who each interviewed 11 GPs. The GP interview guide questions gathered information on their views regarding the use and potential impact on clinical practice of the intervention, specifically the brochure and video materials given to patients. The topics covered included participants' opinions on the facilitators and barriers to implementing the intervention, feedback from patients and colleagues, recommendations for regular use and ideal procedural integration, and the likelihood of future utilization.

Patients' interviews were performed by one researcher (CC). The questions focused on their experiences with the brochure and video, their opinions on the content of those materials, preferences regarding the delivery of the intervention, and suggestions for improving engagement with CRC screening.

In particular, the question posed to the GP in the interview guide was: 'What did your patients think of the brochure and the video?' (see Appendix 1). Similarly, patients were asked: 'What do you think of the brochure?' and 'What do you think of the video?'. These questions helped identify differences or similarities in perceptions regarding the information tools.

## Data analysis

The interviews were recorded either online or using a dictaphone and transcribed using an external transcription company (AMK, France). The transcripts were anonymised, with only the socio-demographic (age, gender) and literacy data (for patients) linked to them. The transcripts were thematically coded and analyzed using Nvivo 14 software. Each interview was double-coded. Two researchers independently coded each GP interview transcript (AB and GC) and two others independently coded the patient interview transcripts (CC and CA) using a framework analytic approach [29,30]. In both cases, a codebook was created containing all the potential codes. Sub-codes were grouped to identify the primary codes relevant to our research questions. The qualitative analysis adopted a reflective approach [31] by attempting to deconstruct the discursive data as objectively as possible in order to limit the impact of their own representations on the thematic analyses. Quotes from the interviews were selected to represent the concepts, ideas, or perspectives most effectively. Those incorporated in this article were deliberately drawn from a diverse range of participants.

## Coding the two sets of interviews

Due to the different sets of interviews conducted and the different objectives, several main codes and sub-codes were identified. For GPs interview analysis, we selected codes that encompassed aspects related to the implementation of the intervention, including the barriers and benefits associated with the use of the brochure and the video, suggestions for their enhancement, facilitators aiding implementation, as well as factors such as time constraints and the diffusion of the intervention.

For the patient interview analysis, codes related to patients' experiences with the use of the brochure and video, their opinions on, and levels of comprehension of, the content of the materials and their suggestions for improving engagement

with CRC screening. The codes related to the barriers and benefits associated with both the use of the brochure and the video were also included. Only the codes we considered most relevant for our cross-analysis with the GP interviews were included in the results.

## Results

### 1. Demographics characteristics of participants

A total of 22 General Practitioners (GPs) and 24 patients participated in the interviews. All the 32 GPs in the intervention arm were initially invited. There were different reasons for non-participation: some GPs had already withdrawn from the trial, some retired or moved to another region.

For patients, the main reasons for not agreeing to take part in the interview were generally a lack of interest in taking part or a lack of time. Of the patients who initially agreed to participate (90 patients), 71% did not respond to requests from the qualitative interview researcher.

The detailed characteristics of the GPs and patients included are shown in Tables 1 and 2.

22 GPs were interviewed of which 13 were women and 9 were men with ages ranging from 29 to 62 years (median age of 35). Five or six GPs from each region participated from multidisciplinary GP practices. Eleven interviews conducted via each method (telephone and videoconference) and the duration of interviews ranged from 30 to 60 minutes.

**Table 1. GPs characteristics.**

| Variables | n | % |
|---|---|---|
| **Location (regions):** | | |
| AURA | 6 | 27% |
| IdF | 6 | 27% |
| Occitanie | 5 | 23% |
| PACA | 5 | 23% |
| **Type of working environment** | | |
| Group practice | 4 | 18% |
| Multi-professional health centre | 12 | 55% |
| Health centre | 6 | 27% |
| **Median age at time of recruitment (IQR)** | | |
| 35 years (30–39) | – | – |
| **Range** | | |
| 29-62 years | – | – |
| **Gender- male** | 9 | 41% |
| **No of years qualified as a GP at the time of recruitment:** | | |
| 0-5 | 8 | 36% |
| 05-10 | 8 | 36% |
| +10 | 6 | 27% |
| **Type of interview** | | |
| Telephone | 11 | 50% |
| Videoconference | 11 | 50% |
| Total | 22 | |

*Regions: AURA -Auvergne-Rhône-Alpes, IdF – Ile de France, PACA - Provence-Alpes-Côte d'Azur*

*µ EDI (European Deprivation Indices) was calculated before inviting GP practices to participate in the DECODE trial. GP practices with a score of 4 or 5 (most deprived) were targeted.*

**Table 2. Patients characteristics.**

| Variables | n | % |
|---|---|---|
| **Location (regions):** | | |
| AURA | 5 | 21% |
| IdF | 6 | 25% |
| Occitanie | 9 | 38% |
| PACA | 5 | 21% |
| **Number of years living in France:** | | |
| 5 to 10 years | 2 | 8% |
| 11 to 39 years | 3 | 13% |
| 40 to 49 years | 1 | 4% |
| Born in France | 18 | 75% |
| **Median age at time of interview (IQR)** | | |
| 62 years (54–67) | – | – |
| **Range** | | |
| 51-73 years | – | – |
| **Gender- male** | 9 | 38% |
| **Native Language** | | |
| French | 18 | 75% |
| Regional languages of Cameroon | 3 | 13% |
| Regional language of northern Algeria | 2 | 8% |
| Regional language of the Ivory Coast | 1 | 4% |
| **Type of interview** | | |
| Telephone | 22 | 92% |
| Videoconference | 2 | 8% |
| **Monthly household income** | | |
| Less than 1200€ | 5 | 21% |
| 1200€ to less than 1800€ | 4 | 17% |
| 1800€ to less than 3000€ | 2 | 8% |
| More than 3000€ | 9 | 38% |
| Did not want to answer | 4 | 17% |
| **Highest educational attainment** | | |
| Schooling interrupted before end of primary years (age 11) | 3 | 13% |
| Secondary education qualification (aged 15) | 2 | 8% |
| Vocational qualification post-secondary school | 5 | 21% |
| Baccalaureate qualification (aged 18) | 0 | 0% |
| Post baccalaureate | 14 | 58% |
| **NVS (Newest Vital Sign) total score [1]** | | |
| Median NVS score (IQR) | *5 (3–6)* | |
| 0-1 | 5 | 21% |
| 2-3 | 3 | 13% |
| 4-6 | 15 | 63% |
| missing | 1 | 4% |
| **SILS (Single Item Literacy Screener) score [2]** | | |
| **"How confident are you in filling out medical forms by yourself?"** | | |
| Always | 10 | 42% |
| Often | 8 | 33% |

*(Continued)*

**Table 2.** (Continued)

| Variables | n | % |
|---|---|---|
| Occasionally | 5 | 21% |
| Rarely | 1 | 4% |
| Never | 0 | 0% |
| *EPICES score* (Evaluation of precariousness and health inequalities in health assessment centers) [3] | | |
| Median EPICES score (IQR) | *17 (6-38)* | |
| 0-30 (living situation considered precarious) | 14 | 58% |
| Total 30-100 (living situation less precarious) | 10 | 42% |
| Total | 24 | |

Regions: AURA -Auvergne-Rhône-Alpes, IdF – Ile de France, PACA - Provence-Alpes-Côte d'Azur.

[1]Mansfield ED et al. Canadian adaptation of the Newest Vital Sign©, a health literacy assessment tool. Public Health Nutr. 2018 21(11):2038-2045.

[2]Powers BJ et al. Can this patient read and understand written health information? JAMA. 2010;304(1):76-84.

[3]EPICES score calculation, Centre Technique d'Appui et de Formation des Centres d'examens de Santé (CETAF), France : https://www.cetaf.fr/wp-content/uploads/2023/07/EPICES_Presentation_20120423.pdf Accessed May 2021

A total of 24 patients were interviewed, 15 women and 9 men, aged from 54 to 67 with a median age of 62 years old. Heterogeneous social (shown by monthly household incomes, highest educational attainment and EPICES scores) and literacy levels (NVS [32] and SILS [33] scores) were represented.

### 2. Patients and GPs prefer video over brochure for clarity, humor, and effective messaging

The video and brochure, both designed to promote CRC screening, were perceived by patients and GPs as serving different roles. The brochure was viewed as a useful reminder for patients to use at home, while the video was regarded as a tutorial-style tool. However, while the video was valued for its clarity and engaging approach, the brochure was often underutilized, with some GPs finding it redundant. This section explores how each tool was received and the reasons behind the clear preference for the video.

a) Video: A Pedagogical and Engaging Tool

The video was widely appreciated by both patients and GPs for its ability to clearly explain the screening process. More than 21 out of 22 GPs considered it an effective communication tool, especially for patients who struggle with reading French.

"S*o, for me, the video is to ensure that the patient sees it 'in real life' in a rather pleasant and modern way without encountering any barriers*." – GP [3]

Patients also emphasized the benefits of the visual format, particularly for demonstrating the test procedure, making it easier to understand.

"*Well, with video, you can... you... you understand because the man is... he's filming himself in action. So it's... it's... it's very meaningful, it's very meaningful. And even if you don't understand all the words, I think that by seeing the person do it, you come to... to understand.*" – Patient [1]

The inclusion of subtitles in the video further enhanced accessibility, catering to individuals with hearing impairments. However, two GPs expressed concerns about the lack of inclusivity in the video's portrayal, emphasizing the need for diverse representation to resonate with a broader audience.

b) Humor as a tool to normalize CRC screening

The humorous tone of the video was widely praised by both patients and GPs for its ability to alleviate anxiety and normalize discussions about CRC. The actor portrayed the process of using the toilet for the test in a humorous way by blending actions, words, and mimics, which amused patients during the display of video, according to the GPs. The video's vivid depiction of stools provided a more approachable angle to a typically uncomfortable topic, as reported by GPs. Furthermore, two-thirds of the patients (n = 16) emphasized the significance of the light-hearted tone adopted in the video. The main character's humorous and de-dramatizing demeanor was appreciated. This approach effectively addressed some patients' fears regarding CRC, a topic often deemed « taboo » or sensitive. By injecting humor, the video successfully normalized the situation, making it more palatable for patients.

*"I found it very approachable and I thought the guy was, he was very simple and he expresses himself very well and in fact, it de-dramatizes a lot. There's even a touch of humor at one point when he's on the throne."- Patient* [15].

This perception was shared by GPs, who noted that humor helped reduce patient stress and make the screening process feel less intimidating. Several GPs reported that patients often laughed or smiled while watching the video, which created a more relaxed atmosphere during consultations.

*"I think it's great when patients laugh while watching it because it's a topic that can be a bit stressful, of course, we're talking about cancer screening, but actually, almost every time, they laugh. [...] I think it makes it less dramatic [...] So, they seemed a bit more... well, a bit reassured [...] to say: 'Well, actually, it's fine, it's really not complicated'." – GP* [3]

c) Brochure: underutilization and misunderstanding

Despite the brochure's potential, GPs mentioned that they often underutilized it. More than two-thirds (n = 14) indicated that they had not used it and had simply handed it over without much conviction. Seven GPs cited their familiarity with the existing brochure in the French screening kit as a reason for this underuse.

*"(...) it's true that I didn't find the brochure very user-friendly. But I'm telling you, I was so used to the other one that maybe I also had a mental block." – GP* [5]

Almost half of GPs (n = 9) expressed the opinion that the study brochure was redundant and 1 GP felt the screening kit brochure was redundant. Consequently, there were suggestions to combine the two brochures into a single, more streamlined resource to reduce paperwork for both patients and practitioners.

*"Combining the two, because it's a lot of paperwork for them, you know." – GP* [7]

From the patients' perspective, the brochure's content was still considered too dense, which tended to water down the key messages:

*"...you can't say it's poorly done, but I find it packed with a lot, a lot of information. For example, I preferred the video a hundred times over this" Patient* [14]

Several misunderstandings were also raised, such as the images of the colon and intestine not being explicit enough, or words that were not understood, such as "literacy", misunderstood by at least two respondents:

*"Everything is complicated, the terms are complicated, so there you go, we really need a lot more diagrams, and bigger diagrams."* Patient [13]

However, among the respondents, 10 said they had used the DECODE brochure to talk about colorectal cancer screening with those around them (family, colleagues), and 13 are planning to do so:

*"I had already had it in my hand [...] I think a year ago, as my mum had just had a positive test result [...] she was really worried [...] we had looked at it together, I had tried to reassure her a little "* Patient [14].

*"Oh yes, of course (regarding sharing the brochure) [...] if, for example, at work, someone was to bring up the subject with me [...] I would easily be able to tell them that there are tools that are very well suited for this and that they can check out."* Patient [12].

d) A preference for the video over the brochure

Most patients preferred the video due to its educational nature and ease of understanding, particularly regarding the self-test procedure. They appreciated the visual demonstration provided by the video, which simplified complex concepts and made the process more accessible:

*"(...) by reading, I find it much less accessible than video"* – Patient [6]

As expressed by almost half of patients (n = 10), communicating orally and mimicking technical and medical information allowed for greater understanding and accessibility of the information.

*"Well, maybe the video will explain what the brochure doesn't, how to properly follow the sampling method. The sampling method, eh, it does the... it's well shown how... how you have to do it simply and quickly "* - Patient [4]

Almost all the GPs (n = 21) echoed patients' preferences for the video, highlighting its ability to facilitate communication, promote understanding, and serve as a memory aid for medical professionals during consultations:

*"The fact that there was also the video, it's true that it was still, let's say, an additional comfort. It allowed me not to forget anything."* – GP [11]

According to half of the GPs (n = 11), the highly visual format of the video, resembling a tutorial, was well-suited not only for "patients struggling with reading French" but also generally appeared suitable for all audiences.
Overall, while both the video and brochure had their merits, the video emerged as the preferred tool for promoting CRC screening due to its clarity, accessibility, humor and ability to engage and reassure patients effectively.

**3. Integration of the DECODE intervention into primary care consultations: duration, challenges, and patient preferences**

a) Consultation context and time constraints

The integration of the DECODE intervention into consultations varied among GPs and circumstances. Factors such as the accumulated delay of consultations throughout the day, and the reasons for the patient's visit, influenced the time dedicated to presenting the intervention. The brochure and video were typically presented during routine or follow-up

consultations, usually towards the end or occasionally before the clinical examination (n = 20). A minority of GPs sometimes offered it in the waiting room (n = 2) before or after the consultation, to save time. The duration of the intervention, both showing the video and providing the brochure, ranged from 5 to 10 minutes, which is significant given that consultations lasted between 15–30 minutes. Challenges arose when consultations were rushed or disrupted, leading to difficulty in prioritizing CRC screening discussions.

*"There are times when I would have wanted to do it, and we had already spent 40 minutes, you think, 'Well, I won't do it today, you know." – GP* [1]

Even though GPs faced these challenges, patients still expressed a clear preference for receiving the DECODE brochure and video during consultations with their GP or another healthcare professional, valuing the opportunity for discussion and finding the presence of a healthcare professional reassuring and valuable. Most patients were not interested in using the QR code for home access before or after the consultation with their GP.

*"If we receive this kind of document by post, it's likely to end up in the garbage can. On the other hand, if a doctor shows them to us, it's much more valuable and I think it's more likely to make an impression." – Patient* [22]

This sense of direct engagement with a healthcare professional was particularly important for patients who found that a conversation helped simplify complex information and ease concerns about screening.

*"... I found it very useful …it makes things a lot easier to understand…. we need someone to talk to us from time to time, even if we have intellectual abilities, to talk to us simply as if almost to a child, to de-dramatize something that isn't so, but that might seem so even to more... more educated people." – Patient* [19]

b) Time as a necessary investment and a key barrier

All GPs emphasized the importance of investing time in explaining screening procedures to patients. They believed that this investment not only enhanced patient understanding and acceptance but also strengthened the patient-healthcare professional relationship. GPs who initially hesitated to allocate time for patient prevention have recognized the value of thorough discussions in facilitating patient involvement and adherence to screening protocols.

*"Yeah, it was kind of time-consuming, but you know, at the same time, I can't say I didn't like it. It's a drawback, but it's linked to any intervention, really. When you want to do it right, it takes time, you know... and for us, we're always dealing with these time management issues every day." – GP* [8]

GPs who previously opted to simply hand out testing kits without dedicating time to explain it to patients have recognized the critical role of patient education and involvement in the screening protocol. Initially concerned that thorough test discussions would overly extend consultations, these practitioners had discovered that integrating test presentation seamlessly into patient encounters was feasible without significantly elongating visits. Their participation in the DECODE trial has led to a shift in perspective, demonstrating that presenting the test need not demand a substantial time commitment.

*"I also think that participating in this study makes you realize that the time it takes to explain to patients how to do colorectal cancer screening isn't necessarily as significant as you might think. So, realizing that through practice probably makes us more likely to... to do it more often later on, you know." – GP* [10]

c) Using the video to optimize time

The brief three-minute video component of the intervention was viewed as an opportunity for time optimization, allowing GPs to attend to other tasks while patients viewed it.

*"Ultimately, the video saves us time because during that time, we can do other things in the patient's file, you know... It's just a matter, I think, of organizing ourselves to... to use it, you know. And knowing that at some point, there will be a time when the patient will be watching the video, and so we will do something else at that time. His prescription, a paper, filling out the paper." – GP* [7]

Moreover, GPs perceived the video as an investment that could potentially save time in the long run by motivating more patients to undergo screening and reducing the need for follow-up reminders.

*"In essence, when the video is useful and justified, it's not a waste of time. On the contrary, it actually saves us time in the long run." – GP* [5]

The importance of the interaction between the patient and GP was highlighted from both perspectives. Despite the time required, the GPs who had adopted it were aware of the importance of talking verbally with patients about screening, and of using their time efficiently in the consultation. However, they also acknowledged that, in a busy practice, time constraints often made it difficult to consistently engage in detailed discussions, forcing them to prioritize certain aspects of care over others.

**4. Improve prevention by disseminating tools and delegating preventive tasks to meet GPs' time constraints and patient preferences**

a) Delegation of preventive tasks to other healthcare professionals

In response to their time constraints and to patient preferences, GPs considered delegating preventive tasks to other healthcare professionals, such as nurses or medical assistants. The influx of patients, the urgency, and the complexity of their situations at times, as well as the high mental burden for caregivers in contact with vulnerable populations, were all arguments put forward to justify the delegation of prevention tasks.

*"Doing it during...during consultations is fine, but often, we'll lack a bit of time and we'll be less thorough than if we had dedicated time. That's why, currently, unable to do it ourselves given the current state of pressure (...) the number of... patients to see, integrating the medical assistant, the nurse practitioner, that can be interesting with dedicated consultations with them (...) ideally, at the moment, that's what I'm thinking about." – GP* [10].

The GPs who had already implemented this system (= 2) noted improved patient care for screening through delegation as dedicated consultations with nurses often resulted in better attendance and outcomes.

*"So, sometimes, often, they don't come just to talk about that. However, when I schedule them with the nurse to discuss this, it's quite surprising, they come more easily. And well, the nurses have time slots... The Asalée nurse [qualified advanced nurse practitioner], she has one-hour slots, you know." – GP* [8]

This GP also observed that the nurse achieved better screening results*: "She's trained in motivational interviewing.*

*So, she had completely mind-blowing success rates for tests, you know. She easily beats me on... on these success rates. So... so in this case, it's an appointment with her, actually. And she shows the video, because I'm overwhelmed. I have a three-week wait."*

According to this GP, there was indeed a connection between the time spent discussing screening and the patient actually taking the test. The presence of a qualified advanced nurse practitioner, the ASALÉE nurse, a specialist of therapeutic education whose role is to facilitate and improve the care of patients suffering from a chronic pathology, certainly enhanced the probability that the patient undertook the screening test.

*I really have the impression that when we have dedicated time, when we take the time, when we watch the video, well, it works... it works better. [...] And in fact, I think there's also a... a proportional effect in terms of time spent and success, you know. So typically, the Asalée nurse, with 45 minutes to an hour, has a better chance than me."- GP [8]*

However, setting up dedicated prevention consultations, whether with a GP or another health professional, was considered ideal but difficult to achieve due to logistical challenges and patient engagement.

*"In an ideal world, I think it would be great if we had the opportunity to present what screening is and the tools in a more collective time (...) it would be even richer because, well already, it allows reaching several people at the same time and it allows for interaction between the people present. (...) After, we know that it's not always easy because to get people to come for collective time, they already have to have an interest in coming (...)". - GP [11]*

b) Expanding awareness and increasing accessibility of CRC screening

To improve prevention, some of the patients had suggested sharing information on CRC to younger people in order to raise awareness as early as possible. Several patients were spontaneously surprised by the age range targeted by the screening procedures [i.e., 50–74], mistakenly thinking that CRC only concerned this age group, which could minimize awareness of the consequences of CRC before age 50 and after age 74.

*"Yes, I wouldn't worry about introducing the tools to... to my wife but also to my children, I think they need to be made aware too." – Patient [12],*

Another patient shared concerns about the upper age limit for screening, questioning why people over 74 were no longer eligible.

*"I was recently concerned by the fact that this test was limited in terms of age, i.e., that from the age of 74... 74, 75, you were no longer eligible..." – Patient [9] NVS score = 3, SILS score = Occasionally, EPICES score = 23,67.*

Furthermore, to increase the visibility and accessibility of the video and brochure, and enable healthcare professionals to integrate the two components into the routine consultation, several patients suggested showing the video and having copies of the brochure in waiting rooms in GPs' offices.

*"And I thought that even in GPs' waiting rooms, why not, uh, make some broadcasts of the video or... of this order as... as general information so that, well... here, "to trivialize this kind of... of examination which allows a follow-up finally of... finally, all that relates to prevention, I find that it is nevertheless a neglected sector by..., the whole medical system, unfortunately." – Patient [6]*

Similarly, another patient emphasized the value of printed materials in waiting rooms, explaining that patients often look at available documents while waiting for their consultation.

*"When we... when... when we're waiting in... when we're waiting in doctors' waiting rooms, personally, I look at all the documents that are available to... to... and I think it [brochure and video] could fit into that framework." – Patient [7]*

In addition to disseminating the tools within the medical environment, other patients and some GPs also suggested disseminating the tools within the media and social networks for greater visibility and accessibility.

The presence of a nurse or other professional figure devoted to prevention could help GPs in taking charge of this part of the consultation. In France, this is a major difference between GPs practicing alone or with associates and multidisciplinary structures.

## Discussion

The objective of this qualitative study was to assess the acceptability and feasibility of a HL intervention designed to enhance CRC screening in disadvantaged areas with regard to both GPs and patients. The primary trial will evaluate the impact of the intervention. Here, we aimed to gain insight into the perceptions of patients and GPs regarding the intervention: a brochure adapted to a low HL and a step-by-step video on how to do CRC screening test.

Our results indicate that both patients and GPs expressed a strong preference for video materials. Even though the brochure was developed with input from individuals from socially disadvantaged backgrounds, it was not fully tailored to their needs. The tutorial-style video offered a visual and verbal format of information, which led to enhanced interaction, enabling patients to follow step-by-step instructions clearly. Our results align with studies showing that videos enhance comprehension among patients with lower HL levels more than written material [34].

Patients were able to clearly understand the advantages and benefits of doing the CRC screening test, dispelling fears and uncertainties. According to Krouse [35], using video in clinical practice has benefits in assisting with decision-making and reducing anxiety. According to the Health Belief Model (HBM) [36,37], individuals are more likely to engage in preventive health behaviors when they perceive a serious health threat, recognize the benefits of taking action, and feel that barriers to action are minimal. In our study, the video appeared to address key perceived barriers to CRC screening by providing a clear, step-by-step demonstration of the test, which likely increased self-efficacy—a crucial determinant of health behavior adoption. Additionally, by normalizing the test through humor, the video may have reduced anxiety and embarrassment, which are often cited as psychological barriers to CRC screening. This aligns with previous findings showing that educational videos increase knowledge and reduce negative attitudes towards CRC screening [38].

GPs emphasized that the video content demystified the screening process, encouraging patients to take proactive steps for their health. GPs reported that patients were more likely to undergo the CRC test after watching the video, even if they initially had hesitations, because of its humorous tone. Humor helped to normalize discussions about CRC screening, making it seem less daunting for patients. The video's humorous approach to stool manipulation can alleviate patients' reluctance or discomfort regarding this aspect of the screening process. Humor has stress-reducing effects by acting as a coping mechanism in uncomfortable situations like cancer screening [39]. Patients tended to pay more attention to it and remembered the information presented, leading to greater awareness of CRC screening importance. The results of a 2021 study analyzing eye movement evidence showed that humorous tobacco prevention ads received longer and more frequent scanning and higher revisitation rates than non-humorous ads [40].

Videos showing real people modeling the desired behavior are more effective for modifying patient behavior than videos with only didactic information [41]. Inclusivity, concerning accessibility, age, class, sex, and ethnicity, have been identified as crucial factors in enhancing the efficacy of these tools. Inclusivity requires diverse representation to engage a wider audience effectively. Additionally, disseminating information about CRC to younger audiences could help raise awareness early. Those of non-white ethnicities must identify with the characters depicted in prevention videos in order to be able to engage with the content effectively. Moreover, it is of the utmost importance that cultural sensitivity be incorporated into the development and delivery of such videos [42]. Older patients (≥65 years old) and individuals from low socioeconomic backgrounds often encounter difficulties using digital devices, which can act as a significant barrier to health interventions [43]. This is why, although integrating the video into the waiting room or providing a QR code on the brochure could be seen as a time-saving solution for GPs, such an approach would not be suitable for our target

population- which includes elderly and socioeconomically disadvantaged patients. Additionally, patients themselves expressed a strong preference for face-to-face interactions over digital-only solutions.

The majority of patients expressed a greater preference for watching the video within the GP practice, underscoring the importance of patient-provider communication and the value of personalized interactions in promoting adherence to CRC screening recommendations. Patients also preferred receiving the brochure and video during a medical consultation with their GP, strengthening and providing credibility to the use and the content of the tools. Patients felt more comfortable discussing sensitive health topics, such as cancer screening, in a clinical setting, where they could ask questions and receive personalized guidance. Our findings align with studies showing that patients with low HL preferred verbal health information, particularly risk information [44] and may face challenges accessing online resources due to limited literacy skills [45]. The absence of physician recommendations is a major barrier, as patients may interpret it as they were not concerned or that the issue was not significant enough to participate in screening [46].

Despite the overall acceptability of the intervention, barriers to implementation were identified. Interviews with GPs revealed that the main barrier they faced in introducing screening for CRC was a lack of time. GPs reported that time constraints often led to screening discussions being deprioritized in favor of more immediate medical concerns, resulting in low engagement with the intervention tools. Some GPs admitted that, due to time pressure, they simply handed the brochure to patients without explaining its content, significantly reducing its effectiveness. To facilitate the normalization of this intervention, and to address GPs' time constraints while honoring patient preferences, new organizational workflows are needed. While the implementation of video displays in GP waiting rooms could be regarded as a time-saving strategy, research indicates that while it may enhance patients' knowledge, it may not be effective in modifying health behaviors [47].

Integrating video presentations into their practice enables GPs to efficiently handle their workload while maintaining high-quality patient care, transforming what might be seen as a time-intensive task into a seamless part of the consultation process. Capitalizing on the video's brevity, GPs achieve a delicate balance between comprehensive patient care and efficient time management. Another possibility is delegating the video presentation task to advanced nursing practitioners or trained healthcare assistants allowing GPs to meet more patients' needs and potentially enhance patient engagement more effectively. The GPs who had implemented this system observed improved patient care through delegation, as dedicated consultations with qualified advanced nurse practitioners often resulted in better attendance and outcomes. Our findings align with several studies conducted in France and other countries that have demonstrated that the integration of GPs and qualified advanced nurse practitioners is associated with enhanced productivity among GPs compared to that observed in single-discipline private practice [48]. One GP noted that nurses, trained in motivational interviewing, achieved impressive success rates for screenings, largely due to the time spent on consultations. Indeed, several studies have demonstrated that motivational interviewing has the potential to enhance the uptake of health screenings [49]. While delegating the task to nurses or medical assistants has shown promising results, this approach requires sufficient staffing and adequate training, which may not be available in all healthcare settings.

Several limitations of this study need to be acknowledged. First, while the DECODE trial's aim was to use the principles of good health literacy to improve participation in the screening programme, and we focused on recruiting GPs in disadvantaged areas where the concentration of patients with low HL is often higher, a notable proportion of respondents had high HL levels. Additionally, it is well-known that individuals belonging to higher social groups tend to participate in studies more than those from disadvantaged backgrounds.

It is possible that the participants' responses were influenced by social desirability bias. The participants may have been inclined to provide answers perceived as socially acceptable, particularly to express more favorable opinions towards the video and brochure than they would have in an informal setting. However, it should be noted that the patients interviewed were quite capable of criticising the brochure produced by the research team, which goes against a very strong social desirability bias.

A self-selection bias may also have influenced our results. Patients and GPs who agreed to take part in the study are potentially more aware of prevention and screening issues than those who refused.

This research was conducted within the specific context of France, which may limit the generalizability of the findings to other cultural settings and healthcare systems. The nested nature of the study within a trial also implies that the sample may not be wholly representative of all GPs or patients. Specifically, just few single-handed GPs were included and they have withdrawn from the study, and most participants operated within multidisciplinary practices. This could limit the generalizability of our findings, as single-handed GPs may face different constraints and working conditions that were not captured in our analysis. Furthermore, there was some confusion among respondents regarding which brochure was being referred to, which could have impacted the clarity and consistency of the responses. In addition, while this study focused on the acceptability of the intervention, it did not assess long-term changes in screening behavior. However, this limitation will be addressed in the primary trial, which was designed to evaluate the actual impact of the intervention on screening uptake up to a year after the recruitment consultation.

Despite these limitations, the study presents several strengths. It is noteworthy that the results align with observations from various Western healthcare systems. The targeted context of demographic areas with notably low CRC screening rates underscores the importance of focusing research in communities that are often overlooked. This approach provided valuable insights into the daily lives and workloads of GPs working in these areas and highlighted strategies for improving screening participation. Additionally, by concentrating on improvements in harder-to-reach areas, the findings suggest that these strategies could also be effective in more affluent regions. Finally, the structured context of the trial facilitated the effective and efficient conduct of this research, enabling a thorough exploration of the factors influencing CRC screening rates and GP workloads in underserved communities.

## Conclusion

By considering the needs and preferences of both patients and GPs, healthcare providers can optimize the delivery of screening interventions and promote patient engagement and empowerment in healthcare decision-making. Our findings underscore the importance of tailoring promotional materials to meet the needs of both patients and healthcare professionals to enhance CRC screening uptake. It highlights the importance of striking a balance between the efficiency of digital tools and maintaining good and trusting relationships at the core of healthcare.

From a practical perspective, our study suggests that video-based interventions could be integrated into primary care settings in a way that respects both the time constraints of GPs and the preferences of patients for in-person communication. To maximize feasibility, healthcare providers could implement structured workflows, such as showing the video during consultations while doing administrative tasks or delegating the screening promotion task to advanced nurse practitioners or trained healthcare assistants, who have demonstrated success in patient engagement.

At a policy level, integrating this video into national CRC screening programs could be an effective way to improve adherence rates, particularly in disadvantaged communities.

Integrating humor into CRC screening promotion can create more engaging and relatable messaging, addressing patients' emotional concerns and encouraging proactive participation. This approach has the potential to improve screening uptake. Delegating preventive tasks such as CRC screening promotion to qualified advanced nurse practitioners could prove to be a promising strategy, leveraging their collaborative integration with GPs to enhance CRC rates.

In addition to those results, the primary trial (NCT04631692) of the study will evaluate the actual impact of the intervention on screening uptake up to a year after the recruitment consultation. Ensuring that CRC screening interventions are tailored, accessible, and engaging is critical to addressing disparities in screening participation.

By leveraging audiovisual tools, humor, and trusted healthcare professionals, health systems can empower patients, reduce screening hesitancy, and ultimately improve early detection and outcomes for colorectal cancer, particularly among disadvantaged populations (Table 3).

**Table 3. Comparison of GP vs. Patient Opinions.**

| Themes | GP' opinions | Patients' opinion |
|---|---|---|
| **Use of the video** | Video considered highly educational and accessible, particularly for patients with reading difficulties Appreciated for making the topic less daunting, facilitating test acceptance | Video perceived as clear, simplifying the test process Humor helps reduce anxiety, making screening more approachable |
| **Use of the brochure** | Often underutilized as it is perceived as redundant | Brochure seen as dense, sometimes misunderstood, but useful for sharing with relatives |
| **Time constraints** | Lack of time during consultations, difficulty in integrating an in-depth discussion on screening | Patients appreciate the time GPs dedicate to explanations |
| **Preference for distribution setting of tools** | Prefer distribution during consultations but sometimes used in the waiting room | Prefer distribution during consultations, with physician guidance |
| **Suggestions for improving implementation** | Proposal to delegate prevention to nurses or medical assistants | Suggestion to broadcast the video in waiting rooms and on social media |

## Characteristics of research team

The principal investigators (PIs) M-A Durand, PhD and A Lamouroux PhD are experienced public health researchers with backgrounds in health psychology, interventions and qualitative methodologies. The GPs interviews were conducted by trained researchers (G Cazorla, MSc and A Boirot, PhD) with the trial team. Patient interviews were conducted by a different researcher (C Casanova, PhD) of the same team, all trained in qualitative methodologies and analysis. No prior relationship between interviewers and patients or GPs were established before the interviews, and the only information given to patients and GP about the interviewer was that she was member of the research team of the study. In addition, the wider trial team supported the recruitment of GPs and patients to the qualitative study including the trial manager (NM Redmond, PhD) and the 4 regional Clinical Research Associates (M Kaou, MSc, Z Doukhi, MSc, M-C Addamiano, PhD, F Couranjou, MSc). Standardized procedures were used to support the recruitment and the wider team aided the qualitative study in data cleaning and data analysis.

## Supporting information

**S1 Fig. Full interview guides used for patients and general practitioners in the qualitative study.**
(DOCX)

**S2 Fig. Consolidated criteria for reporting qualitative research used for this study.**
(PDF)

## Acknowledgments

We would like to warmly thank all the participants of the DECODE trial, GPs and our colleagues at the four Regional Cancer Screening Coordination Centres, and Fecop (Federation of primary care coordinated multi-professional practices) and our other partners who assisted with this trial, without whom, this study would not be possible. We would also like to thank our colleagues A. Martinez, R.Grami and N. Allami who replaced the Clinical Research Associates during periods of absence. We would also like to extend our thanks to all the members of our Community Advisory Board for sharing their expertise and insights and our Steering Committee members for their valuable input and advice.

## Author contributions

**Conceptualization:** Niamh Redmond, Julien Mancini, Marie-Anne Durand, Aurore Lamouroux.

**Data curation:** Maria-Claudia Addamiano, Niamh Redmond, Marie-Anne Durand, Aurore Lamouroux.

**Formal analysis:** Alix Boirot, Maria-Claudia Addamiano, Clémence Casanova, Niamh Redmond, Géraldine Cazorla.

**Funding acquisition:** Julien Mancini, Marie-Anne Durand, Aurore Lamouroux.

**Investigation:** Alix Boirot, Maria-Claudia Addamiano, Clémence Casanova, Niamh Redmond, Géraldine Cazorla, Marie-Anne Durand, Aurore Lamouroux.

**Methodology:** Marie-Anne Durand, Aurore Lamouroux.

**Project administration:** Anne-Marie Schott-Petelaz, Christian Balamou, Julien Mancini, Marie-Anne Durand, Aurore Lamouroux.

**Resources:** Julien Mancini, Marie-Anne Durand.

**Software:** Niamh Redmond.

**Supervision:** Michel Rotily, Anne-Marie Schott-Petelaz, Christian Balamou, Julien Mancini, Cyrille Delpierre, Marie-Anne Durand, Aurore Lamouroux.

**Validation:** Marie-Anne Durand.

**Visualization:** Alix Boirot, Maria-Claudia Addamiano, Clémence Casanova, Niamh Redmond, Géraldine Cazorla.

**Writing – original draft:** Alix Boirot, Maria-Claudia Addamiano, Clémence Casanova, Géraldine Cazorla.

**Writing – review & editing:** Alix Boirot, Maria-Claudia Addamiano, Clémence Casanova, Niamh Redmond, Géraldine Cazorla, Michel Rotily, Anne-Marie Schott-Petelaz, Christian Balamou, Zineb Doukhi, Myriam Kaou, Françoise Couranjou, Julien Mancini, Cyrille Delpierre, Marie-Anne Durand, Aurore Lamouroux.

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
