## [Editor Report · Decision Letter 0]

PONE-D-24-31709Acceptability and implementation potential of a health literacy intervention to increase colorectal cancer screening in deprived areas: a qualitative study of patients and general practitioners participating in a cluster randomized controlled trialPLOS ONE

Dear Dr. Boirot,

Thank you for submitting your manuscript to PLOS ONE. After careful consideration, we feel that it has merit but does not fully meet PLOS ONE’s publication criteria as it currently stands. Therefore, we invite you to submit a revised version of the manuscript that addresses the points raised during the review process.

We look forward to receiving your revised manuscript.

Kind regards,

Ateya Megahed Ibrahim El-eglany

Academic Editor

PLOS ONE

Journal Requirements:

2. In the ethics statement in the Methods, you have specified that verbal consent was obtained. Please provide additional details regarding how this consent was documented and witnessed, and state whether this was approved by the IRB.

3. Thank you for stating the following in the Competing Interests section: “I have read the journal's policy and one author of this manuscript have the following competing interests: MA-D has contributed to the development of Option Grid patient decision aids. EBSCO Information Cervices sells subscription access to Option Grid patient decision aids. She receives consulting income from EBSCO Health, and royalties. No other competing interests declared.”

Additional Editor Comments:

Abstract

1. Expand Background: Briefly introduce why colorectal cancer (CRC) screening is critical, particularly among socially disadvantaged populations.

2. Highlight Methodology: Specify the study design (qualitative, thematic analysis) and how participants were recruited.

3. Clarify Key Findings: Clearly state the level of preference for the video versus the brochure (e.g., percentages or qualitative themes).

4. Strengthen the Conclusion: Suggest how findings may be translated into policy or practice, particularly in improving screening uptake.

Introduction

1. Define the Problem Clearly: Provide more statistics on CRC incidence, mortality, and screening uptake disparities to establish the study’s significance.

2. Link to Health Equity: Discuss the social determinants of health and their role in screening participation, incorporating health literacy and digital access considerations.

3. Intervention Justification: Elaborate on why video interventions might be more effective than brochures, citing relevant behavioral science theories.

4. Hypothesis or Study Objectives: Clearly state the research questions or hypotheses guiding the study.

Methods

1. Clarify Inclusion/Exclusion Criteria: Define how patients were selected, specifying characteristics such as age, previous screening history, or language barriers.

2. Expand on GP Recruitment: Explain how GP practices were approached and how many declined participation.

3. Interview Protocol: Provide more details on the interview guide—what key questions were asked, and how responses were probed for depth?

4. Data Management: Describe how qualitative data were transcribed, coded, and validated (e.g., was double-coding used for reliability?).

5. Ethical Considerations: Clarify how participant confidentiality was ensured and whether incentives were provided.

Results

1. Demographic Breakdown: Include a table summarizing participant characteristics (e.g., age, gender, education, previous screening history).

2. Thematic Structure: Clearly separate results by themes, ensuring that GP and patient perspectives are distinct.

3. Direct Quotes for Depth: Incorporate more illustrative patient and GP quotes to support key themes.

4. Unexpected Findings: Discuss any surprising responses, such as participants who still preferred brochures despite the majority favoring videos.

5. Barriers to Implementation: Describe any challenges participants mentioned in accessing or engaging with the intervention.

Discussion

1. Compare to Existing Literature: Situate findings within the broader research on CRC screening interventions, highlighting similarities and differences.

2. Behavior Change Mechanisms: Explore why the video may have been more effective, referencing behavior change theories (e.g., Health Belief Model, Theory of Planned Behavior).

3. GP Workload Constraints: Discuss how time constraints among GPs might limit intervention feasibility and suggest strategies to overcome this.

4. Digital Divide Issues: Acknowledge potential limitations related to digital literacy and access to video interventions in disadvantaged populations.

5. Cost-effectiveness Consideration: Briefly discuss whether video interventions are cost-effective compared to traditional methods like printed brochures.

Limitations

1. Potential Biases: Address possible social desirability bias in participant responses.

2. Generalisability: Acknowledge that findings may not apply to other healthcare settings or more affluent populations.

3. Short-Term vs. Long-Term Impact: Mention that the study focused on acceptability but did not measure long-term changes in screening behavior.

4. Self-Selection Bias: Discuss whether more health-conscious individuals may have been more likely to participate.

Conclusion

1. Summarize Practical Takeaways: Provide clear recommendations for implementing video-based interventions in primary care.

2. Call for Future Research: Suggest the need for a follow-up study assessing actual screening uptake after exposure to the intervention.

3. Policy Implications: Discuss how health systems can integrate video interventions into routine CRC screening promotion strategies.

4. Final Thought: End with a strong statement on the importance of tailored interventions for improving screening rates among disadvantaged groups.

Additional Recommendations for Tables & Figures

1. Comparison Table: Include a table comparing the key themes between GP and patient responses.

2. Flowchart: If applicable, provide a flowchart of participant recruitment and data collection.

3. Graphical Abstract: Consider adding a simple visual summary of the study’s main findings for better accessibility.

---

## [Author Response · Author response to Decision Letter 1]

21 May 2025

To the Editorial Board of PLOS One

Dear Editors,

Thank you for your feedback on our original article entitled “Acceptability and Implementation Potential of a Health Literacy Intervention to Increase Colorectal Cancer Screening in Deprived Areas” and the opportunity to submit a revised draft of the manuscript to Plos One.

We responded point-by-point below to the reviewers’ comments, detailing changes made to the manuscript. These changes are highlighted in red in the revised version.

We hope these modifications adequately address the issues raised, and we remain at your disposal should further changes need to be made to the manuscript.

Yours faithfully,

Alix Boirot, on behalf of all the authors

Journal Requirements:

2. In the ethics statement in the Methods, you have specified that verbal consent was obtained. Please provide additional details regarding how this consent was documented and witnessed, and state whether this was approved by the IRB.

Answer:

We have added a paragraph explaining the Ethics approval in the method’s section, and we had already explained how GPs recruited patients with verbal consent required, which was approved by the Ethics committee in the section “Patient recruitment”: “As approved by CER, GPs obtained verbal consent from the patient, explained the study to them, and gave them a CRC self-test kit. In the intervention group, the GPs also used their training, the pictorial brochure, and the video to explain how to do the CRC screening test at home. All patients were followed up by telephone after 1 week and then 1 year to determine if they had taken the test (primary outcome) » We have added the following sentence for clarity (p.8, line 190) “At each interaction with a recruited patient, we checked they were still in agreement to consent, giving participants the opportunity to withdraw if they did not want to continue.”

3. Thank you for stating the following in the Competing Interests section: “I have read the journal's policy and one author of this manuscript have the following competing interests: MA-D has contributed to the development of Option Grid patient decision aids. EBSCO Information Cervices sells subscription access to Option Grid patient decision aids. She receives consulting income from EBSCO Health, and royalties. No other competing interests declared.”

Answer: all these statements have been added: “We have read the journal's policy and one author of this manuscript has the following competing interests: Marie-Anne Durand has contributed to the development of Option Grid patient decision aids. EBSCO Information Cervices sells subscription access to Option Grid patient decision aids. She receives consulting income from EBSCO Health, and royalties. This does not alter our adherence to PLOS ONE policies on sharing data and materials. No other competing interests declared.”

Answer: As stated during the submission process, all data are available by request to the research team. Please contact either the corresponding author or the trial manager Dr Niamh M Redmond (niamh-maria.redmond@univ-tlse3.fr) for further information.

Answer: the ethics statement is now in the “methods” section

Answer: At the end of the manuscript there is now a statement about the list of supporting files available.

Additional Editor Comments:

Abstract

Reviewer’s comments:

1. Expand Background: Briefly introduce why colorectal cancer (CRC) screening is critical, particularly among socially disadvantaged populations.

2. Highlight Methodology: Specify the study design (qualitative, thematic analysis) and how participants were recruited.

3. Clarify Key Findings: Clearly state the level of preference for the video versus the brochure (e.g., percentages or qualitative themes).

Response:

- Point 1 : We had already addressed this in the abstract: “In France, despite a free organised screening programme for people aged between 50 and 74, participation rates remain suboptimal. Socioeconomic position and health literacy levels exacerbate the situation, with the lowest screening rates observed in the most socially disadvantaged areas of the country.”

- Point 2: We had already stated this in the abstract (please see bold wording). We added “in the DECODE project” in the abstract for clarity (p.2, line 46):

“We conducted a cross-sectional qualitative study using semi-structured interviews with patients (n=24) and GPs (n=22) who used or participated in the DECODE project intervention. The interviews were conducted by telephone or videoconference, and were analysed thematically using Nvivo software and dual independent coding.”

- Point 3: The result section of the abstract has been rearranged in order to highlight the findings (p.2, line 50):

“95% of GPs expressed a clear preference for the video over the brochure. Patients had varied results with 50% preferring the video, as it demonstrated how to do the test, versus the brochure. The humorous and de-dramatising aspects of the video were the two key factors highlighted by interviewees. However, support from healthcare staff (GPs, nurses, etc.) is still essential, in supporting patients in prevention. This presents a challenge for GPs, who are frequently constrained by time limitations during consultations.”

Reviewer’s comments:

4. Strengthen the Conclusion: Suggest how findings may be translated into policy or practice, particularly in improving screening uptake.

Response:

- We have revised the abstract's conclusion to highlight how our findings can inform policy and practice, particularly in improving CRC screening uptake (p.2, lines 59-62):

“Our findings emphasize the need to tailor promotional materials for both patients and healthcare professionals to improve CCR screening uptake, balancing digital efficiency with maintaining core human relationships in healthcare. Such intervention can be integrated into different workflows. The addition of video into national CRC screening programs might also help. Targeting CRC screening interventions at provider-patient interactions, ensuring they are tailored, accessible, and engaging, is key to reducing disparities.

Introduction

Reviewer’s comments:

1. Define the Problem Clearly: Provide more statistics on CRC incidence, mortality, and screening uptake disparities to establish the study’s significance.

Response:

- The introduction has been expanded with additional references supporting our arguments (p.3, lines 71-76):

“According to the World Health Organization (WHO), since 2020, 1.8 million new cases of colorectal cancer have been recorded worldwide, making it the third most common cancer after lung and breast cancer, and the second leading cause of cancer deaths. This disease predominantly affects Western countries, with particularly high incidence rates observed in Europe, Australia and New Zealand (4).”

Reviewer’s comments:

2. Link to Health Equity: Discuss the social determinants of health and their role in screening participation, incorporating health literacy and digital access considerations.

Response:

- The introduction has been expanded with additional references arguing our statements (p.4, lines 86-96):

“Since the implementation of the national CRC screening programme, participation rates have remained unchanged despite campaigns to increase accessibility to CRC screening kits - available for free in pharmacies or directly online at ameli.fr (12). A recent study (13) demonstrated that in France the most disadvantaged populations (based on European Deprivation Index (EDI) data) had a lower net 5-year survival rate for most cancers and in particular for CRC. These findings align with international data (14). Additionally, social inequalities in health affect the most disadvantaged populations both in terms of higher exposure to risk factors and reduced access to care (15). As well as socioeconomic position, lower health literacy is associated with poor health outcomes, poorer overall health status, higher mortality rates (16), and less engagement with prevention services including CRC screening (17).”

Reviewer’s comments:

3. Intervention Justification: Elaborate on why video interventions might be more effective than brochures, citing relevant behavioral science theories.

Response:

- Our study does not aim to compare the effectiveness of video versus brochure interventions; rather, it examines participants' preferences regarding these formats.

Reviewer’s comments:

4. Hypothesis or Study Objectives: Clearly state the research questions or hypotheses guiding the study.

Response:

- Hypothesis have been enhanced (p4, lines 104-107):

“The involvement of general practitioners (GPs) is essential, as they serve as the primary point of contact for many people. Our hypothesis is that the manner in which the importance of the CRC test is introduced and presented, using clear language plays a crucial role in screening rates, particularly among the most disadvantaged populations.”

Methods

Reviewer’s comments:

1. Clarify Inclusion/Exclusion Criteria: Define how patients were selected, specifying characteristics such as age, previous screening history, or language barriers.

Response:

- The inclusion/exclusion criteria were already explained in the methods section under “Patient recruitment, for more clarity we added some details (p.7, line 171):

(i.e. did not conduct a CRC screening test within the previous two years)

Reviewer’s comments:

2. Expand on GP Recruitment: Explain how GP practices were approached and how many declined participation.

Response:

- A more detailed explanation of GP recruitment has been added (p7, lines 162-164):

“GPs were contacted by email or telephone by the research team and follow-up if necessary. We used our research and community contacts to reach other GPs via ‘snowballing’ to invite other GPs to participate”.

- We do not have data on how many declined as many just did not respond to emails or telephone calls.

Reviewer’s comments:

3. Interview Protocol: Provide more details on the interview guide—what key questions were asked, and how responses were probed for depth?

Response:

- We have added a section that explains the key questions from the interviews (p9 lines 215-218):

“In particular, the question posed to the GP in the interview guide/GP “what did your patients think of the brochure and the video? “ (see Appendix 1) and the questions posed to the patients in the patient interview guide “What do you think of the brochure?“ and “What do you think of the video?“ (see Appendix 1) allowed us to identify any differences or similarities in perceptions regarding the information tools.”

Reviewer’s comments:

4. Data Management: Describe how qualitative data were transcribed, coded, and validated (e.g., was double-coding used for reliability?).

Response:

- This information has already been reported in the section 'Data analysis' data analysis”, however for more clarity we add “each interview was double-coded” before the procedure explanation p9 line 224.

Reviewer’s comments:

5. Ethical Considerations: Clarify how participant confidentiality was ensured and whether incentives were provided.

Response:

- We added a statement in order to describe more in details how confidentiality was ensured (p9, lines 221-222):

The transcripts were anonymised, with only the socio-demographic (age, gender) and literacy data (for patients) linked to them.

Results

Reviewer’s comment:

1. Demographic Breakdown: Include a table summarizing participant characteristics (e.g., age, gender, education, previous screening history).

Response:

- This information was already included in the original submission. To ensure clarity, we have reformatted the tables for improved readability (Tables 1 and 2 in the revised manuscript). The data concerning previous screening history was not available.

Reviewer’s comment:

2. Thematic Structure: Clearly separate results by themes, ensuring that GP and patient perspectives are distinct.

Response:

- We have added thematic subsections in the results section to clarify the presentation of the results as suggested:

2. Patients and GPs prefer video over brochure for clarity, humor, and effective messaging

a) Video: A Pedagogical and Engaging Tool

b) Humor as a tool to normalize CRC screening

c) Brochure: underutilization and misunderstanding

d) A preference for the video over the brochure

3. Integration of the DECODE intervention into primary care consultations: duration, challenges, and patient preferences

a) Consultation context and time constraints

b) Time as a necessary investment and a key barrier

c) Using the video to optimize time

4. Improve prevention by disseminating tools and delegating preventive tasks to meet GPs’ time constraints and patient preferences

a) Delegation of preventive tasks to other healthcare professionals

b) Expanding awareness and increasing accessibility of CRC screening

- Additionally, we included introductory and transition sentences to enhance clarity and ensure a clear distinction between patient and GP pers

---

## [Decision Letter · Decision Letter 1]

Acceptability and implementation potential of a health literacy intervention to increase colorectal cancer screening in deprived areas: a qualitative study of patients and general practitioners participating in a cluster randomized controlled trial

PONE-D-24-31709R1

Dear Dr. Alix,

We’re pleased to inform you that your manuscript has been judged scientifically suitable for publication and will be formally accepted for publication once it meets all outstanding technical requirements.

Kind regards,

Sameen Abbas

Academic Editor

PLOS ONE

Additional Editor Comments (optional):

improve title as per reviewer's comment

Reviewers' comments:

Reviewer's Responses to Questions

**Comments to the Author**

1. If the authors have adequately addressed your comments raised in a previous round of review and you feel that this manuscript is now acceptable for publication, you may indicate that here to bypass the “Comments to the Author” section, enter your conflict of interest statement in the “Confidential to Editor” section, and submit your "Accept" recommendation.

Reviewer #1: All comments have been addressed

Reviewer #2: All comments have been addressed

2. Is the manuscript technically sound, and do the data support the conclusions?

Reviewer #1: Yes

Reviewer #2: Yes

3. Has the statistical analysis been performed appropriately and rigorously? 

Reviewer #1: N/A

Reviewer #2: Yes

4. Have the authors made all data underlying the findings in their manuscript fully available?

Reviewer #1: Yes

Reviewer #2: Yes

5. Is the manuscript presented in an intelligible fashion and written in standard English?

Reviewer #1: Yes

Reviewer #2: Yes

6. Review Comments to the Author

Reviewer #1: The authors addressed the reviewers comments. I believe the manuscript is in a very good shape, now.

Reviewer #2: It is better to rewrite the tittle of the article to concise it (e.g. omit last part: "participating in a cluster randomized controlled trial"

7. PLOS authors have the option to publish the peer review history of their article (what does this mean? ). If published, this will include your full peer review and any attached files.

**Do you want your identity to be public for this peer review?** For information about this choice, including consent withdrawal, please see our Privacy Policy .

Reviewer #1: **Yes: ** Abedalrhman Alkhateeb

Reviewer #2: No
